# Learning a vector field from snapshots of unidentified particles rather than particle trajectories

**Yunyi Shen**[1,*]**, Renato Berlinghieri**[1,*]**, Tamara Broderick**[1]
[1]Massachusetts Institute of Technology
`{yshen99, renb, tbroderick}@mit.edu`

## Abstract

Practitioners frequently aim to infer dynamical system behaviors using snapshots at certain time points. For instance, in single-cell sequencing, to sequence a cell we must destroy it. So we cannot access a full trajectory of the behaviors of a cell, but we can access a snapshot sample. While stochastic differential equations (SDEs) are commonly used to analyze systems with full trajectory access, the availability of only sparse time samples without individual trajectory data makes traditional SDE learning methods inapplicable. Recent works in the deep learning community have explored using Schrödinger bridges for dynamics estimation from such data. However, these methods are primarily tailored for interpolating between two time points and struggle when asked to infer the underlying dynamics that generate all observed data from multiple snapshots. In particular, a naive extension to multiple points performs piecewise perfect interpolation without considering the collective information from all snapshots. In contrast, we propose a new method that leverages an iterative projection mechanism inspired by Schrödinger bridges. Our method does not require that the inferred dynamics precisely match every snapshot, offering a substantial advantage in practical applications where perfect data alignment is rare. By incorporating information from the entirety of the dataset, our model provides a more robust and flexible framework for dynamics inference. We test our method using well-known simulated parametric models from systems biology and ecology.

## 1 Introduction

Often practitioners are interested in forecasting dynamical systems from snapshots at certain time points without access to individual trajectories. For instance, suppose we are interested in understanding the dynamics of mRNA concentration in a population of cells to develop new cancer treatments. We can model the dynamics of mRNA concentration in each cell using a stochastic differential equation (SDE). However, we cannot measure mRNA concentration in continuous time but only at several snapshots. Moreover, because a scientist must destroy a cell to measure its mRNA concentration, we cannot track the trajectory of any *individual* cell. In other words, we have access only to marginal samples of the mRNA concentration at a few snapshots.

While there is extensive theory on SDEs from trajectories densely sampled in time, these methods are not applicable to the kind of data in the mRNA example. In contrast, recent works in the deep learning community have demonstrated the potential of using Schrödinger bridges (SBs) to estimate a system's dynamics in these settings (Wang, Jennings, and Gong, 2023; Vargas, Thodoroff, Lamacraft, and Lawrence, 2021; Lavenant, Zhang, Kim, and Schiebinger, 2021). However, this line of work focuses on interpolating dynamics between two time points and does not address the case where we have access to multiple time snapshots. While it is possible to treat a time series of snapshots as a series of sequential pairs, this approach can lead to neglecting long-term dependencies, seasonalities, and cyclic patterns. Furthermore, SBs require the learned dynamics to satisfy hard constraints of what the distribution of the trajectories should look like at given time steps.

---

*Authors contributed equally

This is not ideal in our setting for two main reasons. (1) We would like to have the flexibility to accommodate deviations from exact marginal distributions, which is often desirable in real-world scenarios. (2) We care about learning dynamics to reflect the evolution of the system over the entire time course, not just piecewise interpolation between each pair of consecutive time points.

Because of these limitations of existing methods, we propose a new method to learn dynamics given only sparse time snapshots of data without individual trajectories. Our method focuses on identifying the dynamic model within a specified parametric family (e.g. adhering to inherent biases such as physical constraints) that is *closest* to *all* the observed marginals, with the flexibility of not having to pass exactly through all of them. Our method iterates between two steps. (1) Learn a piecewise SB to interpolate the unobserved trajectories spanning the entire time course guided by our current best guess of the underlying dynamics. Essentially, this phase uses our initial model to fill in the gaps between observed data points, sketching a continuous trajectory that aligns with the known snapshots. (2) Use the learnt SB to refine our best guess (in the model family) for the underlying dynamics. This step doesn't strictly require the model to match every observed data point perfectly. Instead, it ensures that the refined model is the best at representing the marginal samples within the pre-specified parametric family. This balance allows us to maintain a degree of flexibility, avoiding the trap of overfitting to specific data points while still adhering to our fundamental understanding of the system.

We illustrate our proposal on two well-known simulated parametric models from systems biology (the repressilator system, Nakajima et al. (2005)) and ecology (Lotka-Volterra, Goel et al. (1971)). We show that our method accurately reconstructs the underlying dynamics, and has superior performance compared to a simple parametric generalization of the SB problem to multiple time steps.

## 2 SETUP

In this section, we introduce our problem and the main challenges we aim to address. Following the biology example introduced above, suppose we are interested in understanding the dynamics of mRNA concentration in a population of cells. At time step $t_i, i = 1, \ldots, I$, we observe $N_{t_i}$ cells, indexed by $n_i$, $n_i = 1, \ldots, N_{t_i}$, and measure their mRNA concentration $Y_{t_i}^{n_i}$. Unfortunately, whenever we obtain such a measurement for a single cell, the cell dies. Therefore we can obtain mRNA concentrations for the same cell only once. Although we cannot observe it, we assume each cell would have a continuous time trajectory of its mRNA concentration if the cell were not destroyed. We let $X_t^{n_i}$ denote the trajectory for the cell indexed by $n_i$ as a (continuous) function of time $t$, and we assume we have access to one observation per trajectory, $Y_{t_i}^{n_i} = X_{t=t_i}^{n_i}$. In total, we have $\sum_{i=1}^{I} N_{t_i}$ samples, and each sample has a corresponding latent continuous time trajectory. We assume the time steps are unique and increasing, i.e., $t_1 < t_2 < \cdots < t_I$, but need not be equally spaced. That is, we can have $t_{i+1} - t_i \neq t_i - t_{i-1}$ for some $i = 2, \ldots, I - 1$.

For each particle $n_i$, we model the latent trajectory using an SDE driven by Brownian motion $W_t$:

$$\mathrm{d}X_t^{n_i} = b_0(X_t^{n_i}, t)\mathrm{d}t + \sqrt{\gamma}\mathrm{d}W_t, \quad X_{t=0}^{n_i} \sim \pi_0. \tag{1}$$

We assume that volatility $\gamma$ is known, drift $b_0(\cdot, \cdot)$ is unknown, and we have observed samples of the initial distribution $\pi_0$. See Chapter 3 in Pavliotis (2016) for more details on this SDE formulation. The goal of this work is to learn the underlying dynamic drift, $b_0(\cdot, \cdot)$, from the samples $Y_{t_i}^{n_i}$, $t_i = 1, \ldots, I, n_i = 1, \ldots, N_{t_i}$. This problem is challenging: since each trajectory is evaluated only a single time, we are not able to monitor individual trajectories. This issue makes it hard to apply the classical methods for likelihood estimation with discrete time observations (which typically involve approximating the corresponding Markov kernels as described by e.g., Lo (1988)), and so we propose a new approach to solve this problem.

## 3 OUR METHOD

To estimate the drift function $b(X_t, t)$, we choose to establish a loss function that quantifies the divergence between observed samples $Y_{t_i}^{n_i}$, $t_i = 1, \ldots, I$, $n_i = 1, \ldots, N_{t_i}$, and the trajectories obtained by an alternative SDE characterized by eq. (1), with drift $b$ instead of $b_0$. The best approximation $\hat{b}$ of $b_0(\cdot, \cdot)$ would then be found by minimizing this loss with respect to $b$. Likelihood is

often used as a loss when trajectories are accessible (Lo, 1988). So in principle one could solve the Fokker-Planck equation associated to the SDE with drift $b$ to obtain an equation to evaluate the likelihood function (see Chapter 4 in Pavliotis (2016) for a review on the Fokker-Planck equation). However, this might not be feasible, because during optimization we would need to solve a different partial differential equation for each value of $b$. Schrödinger bridges (SBs, Wang et al. (2023); Vargas et al. (2021); Lavenant et al. (2021)) can help us overcome this issue. The goal of the SB problem is to find a pair of forward-backward SDEs that interpolate between two distributions, ensuring that the learned dynamics are as close as possible — in a Kullback–Leibler divergence ($D_{\mathrm{KL}}$) sense — to a predefined reference dynamics, usually a Brownian motion. In our work, we are interested in a straightforward generalization of the SB problem to multiple time steps (Vargas et al., 2021; Lavenant et al., 2021), which can be adapted to our setting by defining the following optimization problem

$$\arg\min_{q \in \mathcal{D}} D_{\mathrm{KL}}(q||p_b), \quad \mathcal{D} = \{q : q_0 = \pi_0, \ldots, q_{t_I} = \pi_{t_I}\} \tag{2}$$

where the reference measure is $p_b$, the measure associated to the SDE with drift $b$. SB has been proven to be successful in applications with two end points, such as deep generative models (De Bortoli et al., 2021; Wang et al., 2021). Even though eq. (2) appears to leverage information from all snapshots jointly, Lavenant et al. (2021) showed that — under very mild conditions — this SB problem is equivalent to a collection of $I - 1$ separate SB problems, each between two adjacent time steps. That is, this method is essentially performing piecewise interpolation between each pair of consecutive time points. For this reason, the learned dynamics might not be able to capture long-term dependencies, seasonalities, and cyclic patterns, and might not adequately handle noisy data. Making an analogy, it is as if we were trying to solve a regression problem in 2D (e.g., predict cell weight from cell diameter) and for some specific values of the independent variable (cell diameter), the regression function would be constrained to pass through all the observed points. We expect piecewise interpolation to perform poorly in many problems, such as the mRNA example, where (1) we believe there is some structure in the data that can be leveraged (e.g. seasonalities, physical constraints) and (2) due to noisy data and imperfectly specified models, we do not expect the underlying dynamics to pass perfectly through every observed data point.

To overcome this issue, we propose to iterate the following steps: we use the solution to the SB problem to obtain an SDE that we can sample from, and then we use these samples to refine our best guess for the drift $b$. In this refining step, we suggest to find the best drift $b$ in some parametric model family $\mathcal{F}$ by optimizing the second argument of the KL divergence. Concretely, we can write our proposed estimator $\hat{b}$ of $b$ as

$$\hat{b} := \arg\min_{b \in \mathcal{F}} \min_{q \in \mathcal{D}} D_{\mathrm{KL}}(q||p_b), \quad \mathcal{D} = \{q : q_0 = \pi_0, \ldots, q_{t_I} = \pi_{t_I}\} \tag{3}$$

where the distribution constraints $\pi_{t_i}$ are the distributions of the true dynamics (solving the true Fokker-Planck equations) and are accessible only through samples, similar to a typical SB in generative modeling (Vargas et al., 2021). We thereby obtain an estimator that (1) exploits the structure of the model family and (2) has the flexibility of not having to perfectly interpolate each marginal. In the 2D regression analogy, we are no longer forcing the regression function to pass through every observed point, but rather to be close to them while also being smooth and flexible, giving more importance to general trends in the data, rather than the specific values of the observed marginals.

In practice, to solve our optimization problem, we propose an iterative algorithm (Algorithm 1 in appendix B) that alternates between (i) solving the SB problem for the current drift (`Forward-Backward SB`), (ii) generating samples with the solutions of the SB problem (starting from the observed $\boldsymbol{Y}_{t_i}^{n_i}$ and going backward and forward in time with `backwardSDEs` and `forwardSDEs`, respectively), and (iii) updating the drift $b$ given these new samples from the SB-learned SDEs (`DriftFit`). Although in theory one could have full generality for the form of the drift $b$ (i.e. $\mathcal{F}$ can be a very large family of functions), restricting the form of possible model family is needed in practice for identifiability (see Section D). Finally, observe that to have flexibility when solving the SB problem we follow the nonparametric Gaussian process method by Vargas et al. (2021) in `Forward-Backward SB`. The last `DriftFit` can similarly be made nonparametric with another Gaussian process prior or parametric with a neural network.

## 4 EXPERIMENTS

In this section, we evaluate our model on two parametric examples important in systems biology and ecology. While our proposed method does not require the drift function $b(\boldsymbol{X}_t, t)$ (and its associated model family) to be time-homogeneous, we focus our experiments on time-homogeneous drifts (i.e., drifts of the form $b_0(\boldsymbol{X}_t)$). Time-homogeneous drifts can be interpreted as a constant regulatory mechanism among genes or species, making them simpler to understand from a scientific perspective. We compare to two other possible methods. (1) We consider a vanilla application of SB; that is, we consider learning an SDE separately for each pair of adjacent time steps. (2) We consider a simple generalization of Vargas et al. (2021) that fits a parametric forward drift using all time steps (*global-forward-SB*, see appendix B). We show that our method accurately reconstructs the underlying dynamics, and has superior performance compared to the baselines.

**Metrics of success.** Both our method and global-forward-SB can be seen as directly reporting an estimate of the time-homogeneous drift $b_0(\boldsymbol{X}_t)$. We can therefore directly compare the results of these methods to the ground-truth drift on a lattice both visually and via a summary of error; for the summary, we report the mean squared error (MSE) on the lattice. By contrast, the vanilla SB estimates $b_0(\boldsymbol{X}_t)$ separately between every two time steps; indeed forming a single estimator across time steps in the time-homogeneous case can be seen as a contribution of our work. Here we give a visual summary and report the MSE that results from applying vanilla SB for a selection of pairs of time steps. In particular, since we have 10 different time points in each of the experiments below, we report vanilla SB for four time ranges: the first and last time ranges and a roughly evenly spaced selection of internal time ranges.

### 4.1 REPRESSILATOR

In this experiment, we generated synthetic data based on a model that captures the circadian rhythm in cyanobacteria, as detailed in (Nakajima et al., 2005). This system is described by a set of three coupled SDEs, modeling mRNA levels of three genes that cyclically suppress each other's synthesis (see appendix E.1 for more details). For simplicity, we assume that the system is homogeneous in time. The goal is to reconstruct the underlying vector field using the generated samples with no individual trajectories. We test the models on a lattice of $10 \times 10 \times 10$ points in the domain $[0, 7] \times [0, 7] \times [0, 7]$.

**Results.** We report both MSE on the lattice as well as visual summaries; see fig. 1 for the results of our method and global-forward-SB, and see fig. 3 in appendix E for the results of vanilla SB. We observe that our method can use the information from all samples to find an accurate global fit, as clearly shown by the visual similarity to the ground truth and lower mean squared error (MSE) on the lattice (0.807). The global-forward-SB method, on the other hand, predicts a vector field of smaller magnitude, potentially because the reference measure serves as a regularization, and the MSE is higher (9.998). In appendix E, fig. 3, we see that the vanilla SB approach fails to capture the global dynamics. The two time steps used in each instance of vanilla SB can be expected to provide less information about the general system than the full set of time steps used by the other methods. As expected, we see that, for each pair of time steps, the reconstructed field is much worse than the one learnt with the other two approaches. In terms of MSEs, we see that for each experiment the vanilla-SB MSE is higher than the ones obtained by the other two methods; we find vanilla-SB MSEs of 17.8, 17.2, 16.7 and 16.6, respectively. Our results suggest that, in this case, the global information is crucial for the reconstruction of the vector field.

### 4.2 LOTKA-VOLTERRA.

In our second experiment, we generated data from a stochastic Lotka-Volterra predator-prey model; see appendix E for more details. We include this experiment as a classic example of a parametric dynamical system.

**Results.** See fig. 4 in appendix E for the results of our method and global-forward-SB. See fig. 5 in appendix E for the results of vanilla SB. In this case, our method accurately reconstructs the underlying dynamics, with a MSE of 0.05. The global-forward-SB method also produces a predicted

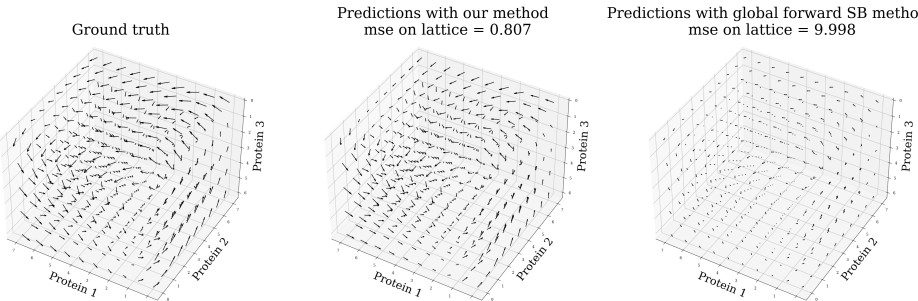

Figure 1: Vector field reconstruction in the repressilator experiment. Left: the ground truth vector field generating data. Mid: reconstruction using our method. Right: reconstruction using global forward SB method. Our method's MSE on the lattice is 0.807. The alternative's MSE is 9.9.

vector field which is very similar to the ground truth, but with a higher MSE (1.265). The vanilla SB approach still fails to capture the global dynamics. as in the repressilator example; we see this discrepancy both visually in the figures as well as in the MSE summaries. The MSEs for each instance of vanilla SB are very large compared to the ones obtained by the other two methods; in particular, we find MSEs of 11.4, 11.8, 11.8 and 11.5, respectively.

## 5 DISCUSSION

We introduced an iterative approach for learning an unknown SDE using only marginal samples, without access to individual trajectories. We show that our method provides superior performance in two parametric experiments with biological significance. Nonetheless, challenges remain. Marginal samples may not always provide sufficient information for accurate system identification (see Section D for a more detailed discussion on identifiability). Moreover, our approach presumes observations are free from observational noise (that is, noise not coming from the SDE but from other sources, e.g. measurement errors). This is often a restrictive assumption in practice. To overcome this limitation, our method could be adapted to include observational noise, following a strategy employed in Wang et al. (2023). Finally, we would like to extend our method to make forecasts, i.e. predict the distribution of latent trajectories at future time points. We believe that our method can be extended to address these challenges, and we hope to explore these directions in future work.

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

## A   APPENDIX

## B   IMPLEMENTATION DETAILS

In this section we discuss the implementation details of our method and the competing SB with global forward drift.

### B.1   OUR METHOD

We present our algorithm in Algorithm 1. We use the method proposed by Vargas et al. (2021) to solve the SB problem, which is based on Gaussian processes (with squared exponential kernel in this case). For `DriftFit`, we fit the drift function using the autoregressive objective function, i.e. we discretize the SDE using the Euler-Maruyama scheme, which is an autoregressive process, and fit a regression model parameterized by the drift function using the output.

---

**Algorithm 1:** Our method with iterative projections

**Data:** marginal samples $\{Y_{t_i}^{n_i}\}_{n,i}$ $i = 1, \ldots, I$, $j_i = 1, \ldots, N_i$

**Result:** drift estimate $\hat{b}$

$\hat{b} \leftarrow 0$ ;                    /* Start from some initial drift guess */

**while** *not converge* **do**

  **for** $i = 1, \ldots, I - 1$ **do**

    $b_i^{\text{forward}}, b_i^{\text{backward}} \leftarrow$ Forward-Backward SB$(Y_{t_i}^{n_i}, Y_{t_{i+1}}^{n_{i+1}} \| \hat{b})$ ;        /* solve SB problem with prior SDE being the current estimated drift */

  **end**

  **for** *all $n_i$* **do**

    $X_{0 \leq t \leq t_i}^{n_i} = $ backwardSDEs$(b_1^{\text{backward}}, \ldots, b_i^{\text{backward}}, Y_{t_i}^{n_i})$ ; /* simulate backward with learned backward SDEs for time before sampling */

    $X_{t_i < t \leq t_I}^{n_i} = $ forwardSDEs$(b_{i+1}^{\text{forward}}, \ldots, b_{I-1}^{\text{forward}}, Y_{t_i}^{n_i})$ ;      /* simulate forward with learned forward SDEs for time after sampling */

  **end**

  $\hat{b} \leftarrow$ DriftFit$(X_{t_t}^{n_i})$ ; /* Fit drift with interpolated trajectories */

**end**

---

### B.2   SB WITH GLOBAL FORWARD DRIFT

The main competing method is a generalization of the method by Vargas et al. (2021) to multiple time steps. The main features of this generalization are that (i) the forward drift part of the SB is shared among all time intervals and (ii) the forward drift is parameterized. To reflect the fact that we are globally sharing a parametric forward SDE, we call this method *global-forward-SB*. The method is described in Algorithm 2.

---

**Algorithm 2:** Global Forward SB

---

**Data:** marginal samples $\{Y_{t_i}^{n_i}\}_{n,i}$ $i = 1, \ldots, I$, $j_i = 1, \ldots, N_i$, reference drift $b$
**Result:** drift estimate $\hat{b}$
$\hat{b} \leftarrow 0$ ;                    /* Start from some initial drift guess */
**for** $i = 1, \ldots, I - 1$ **do**
$\quad$ $Z_{[t_i, t_{i+1}]} \leftarrow$ SDEsolver$(b, Y_{t_i})$
$\quad$ $b_i^{\text{backward}} \leftarrow$ GPDriftFit$(Z_{[t_{i+1}, t_i]})$ ;    /* Initialize the flexible backward
$\quad$ drift */
**end**
**while** *not converge* **do**
$\quad$ **for** $i = 1, \ldots, I - 1$ **do**
$\quad\quad$ $Z_{[t_{i+1}, t_i]} \leftarrow$ SDEsolver$(Y_{t_{i+1}}, b_i^{\text{backward}})$ ; /* solve SB problem with prior
$\quad\quad$ SDE being the current estimated drift */
$\quad$ **end**
$\quad$ $\hat{b} \leftarrow$ DriftFit$(\{Z_{[t_i, t_{i+1}]}\})$ ;    /* Fit drift with backward interpolated
$\quad$ trajectories */
**end**

---

## C  MULTIMARGINAL SCHRÖDINGER BRIDGE

In this section we provide a short review on solving the SB problem with multiple marginals. Indeed, the most classic SB (that achieved great success for example as a deep generative model) only has a pair of end points while we have $I$. The simplest way to solve this problem is to solve a sequence of SB problems between consecutive pairs of marginals, and then the general forward-backward SDE can be obtained by concatenating the forward SDEs and backward SDEs at the corresponding time intervals. This naive approach is theoretically motivated by the work of Lavenant et al. (2021) who showed that if we slightly restrict $\mathcal{D}$ to be Markov, the KL between two measures satisfying all $I$ marginal constraints is lower bounded by the sum of pairwise KL between measures satisfying the pairwise marginal constraints.

**Lemma 1 (Proposition D.1 of Lavenant et al. (2021))** *For a general reference measure $\mathbb{Q}_b^\gamma$*

$$D_{\text{KL}}((\mathbb{Q})_{t_1, \ldots, t_I} || (\mathbb{Q}_b^\gamma)_{t_1, \ldots, t_I}) \geq D_{\text{KL}}((\mathbb{Q})_{t_1, t_2} || (\mathbb{Q}_b^\gamma)_{t_1, t_2})$$
$$+ \sum_{i=2}^{T-1} \left[ D_{\text{KL}}((\mathbb{Q})_{t_i, t_{i+1}} || (\mathbb{Q}_b^\gamma)_{t_i, t_{i+1}}) - D_{\text{KL}}((\mathbb{Q})_{t_i} || (\mathbb{Q}_b^\gamma)_{t_i}) \right]$$

*Inequality attend equal when $(\mathbb{Q})_{t_1, \ldots, t_I}$ is Markov.*

Because of this lemma, when solving the SB problem with more than two marginal constraints, we can find the SBs between consecutive pairs of marginals and then concatenate the forward SDEs and backward SDEs at the corresponding time intervals. As discussed in the main text, this approach can be problematic, because it can lead to neglecting long-term dependencies, seasonalities, and cyclic patterns.

## D  IDENTIFIABILITY CONCERNS

In settings like ours, marginal samples may not provide sufficient information to discern the underlying system. One case this might happen is when the system is at equilibrium to start with, e.g., our initial distribution is the invariant measure of the system (if it exists). However, this is not the only case and could happen even the system is not at equilibrium or even without invariant measure. For the former, consider a scenario where the drift is the sum of a rotationally symmetric gradient field and a constant-curl rotation, with the initial sample distribution also being rotationally symmetric. In this case, the marginal distribution would maintain rotational symmetry at each subsequent time step, irrespective of the constant-curl rotation's angular velocity. This means that merely by

observing marginal samples, we might fail to detect the rotation parameter that controls angular velocity.

One particularly case of a rotational invariant vector filed is a simple vector field with constant curl with Gaussian initial distribution. Formally in the settings of eq. (1), consider a simple drift parameterized by $\alpha \in \mathbb{R}$, $b(\boldsymbol{x}) := [\alpha x_2, -\alpha x_1]^\top$, a simple vector field with constant curl. When $\alpha = 0$ the dynamics is driven just by the Brownian motion. Suppose $\pi_0$ is an isotropic normal, $\pi_0 \sim \mathcal{N}(0, \beta I_2)$ for some $\beta$ (and $I_2$ denoting the 2D identity matrix). For a given constant volatility $\gamma$, when $\alpha = 0$, the distribution of the particles at any time step is always isotropic normal.

Indeed, if we consider the general form of the Fokker-Planck equation, this can be written as

$$\frac{\partial p(\boldsymbol{x},t)}{\partial t} = -\nabla \cdot (b(\boldsymbol{x})p(\boldsymbol{x},t)) + \gamma\nabla^2 p(\boldsymbol{x},t)$$
$$= -p(\boldsymbol{x},t)(\nabla \cdot b(\boldsymbol{x})) - \nabla p(\boldsymbol{x},t) \cdot b(\boldsymbol{x}) + \gamma\nabla^2 p(\boldsymbol{x},t)$$
$$= -\nabla p(\boldsymbol{x},t) \cdot b(\boldsymbol{x}) + \gamma\nabla^2 p(\boldsymbol{x},t)$$

If we adapt this to isotropic Gaussians, we have $\nabla p(\boldsymbol{x},t) \cdot b(\boldsymbol{x}) = 0$, and therefore we only end up with the diffusion term for the Brownian motion, meaning that the distribution of the particles at any time step will always remain isotropic Gaussian with increasing variance.

In general, the identifiability of the drift is an interesting question for future research. We believe that the identifiability literature of partial differential equations (especially Fokker-Planck equations) should be useful in this regard.

## E  EXPERIMENTS

In this section we introduce in more detail the two parametric experiments we used to evaluate our method. We also provide additional figures to compare our method with the competing SB with global forward drift, and the vanilla SB method that solves the SB problem independently for each pair of time steps.

### E.1  REPRESSILATOR

**Experiment setup.** The repressilator is a synthetic genetic regulatory network that functions as a biological oscillator, or a genetic clock. It was designed to exhibit regular, sustained oscillations in the concentration of its components. The repressilator system consists of a network of three genes that inhibit each other in a cyclic manner: each gene produces a protein that represses the next gene in the loop, with the last one repressing the first, forming a feedback loop.

We can model the dynamics of the repressilator using the following SDEs:

$$dX_1 = \frac{\beta}{1 + (X_3/k)^n} - \gamma X_1 + 0.1 dW_1$$
$$dX_2 = \frac{\beta}{1 + (X_1/k)^n} - \gamma X_2 + 0.1 dW_2$$
$$dX_3 = \frac{\beta}{1 + (X_2/k)^n} - \gamma X_3 + 0.1 dW_3$$

where $[dW_1, dW_2, dW_3]$ is a 3D Brownian motion, and the repressing behavior is quite clear from the drift equations. To obtain data, we fix the following parameters: $\beta = 10, n = 3, k = 1, \gamma = 1$.

We start the dynamics with initial distribution $X_1, X_2 \sim U(1, 1.1)$ and $X_3 \sim U(2, 2.1)$. We simulate the SDEs for 10 instants of time. At each time step, we take 20 samples. We use Euler-Maruyama method to obtain the numeric solutions.

**Results.** In fig. 2, we show the results of the vector field reconstruction in the three protein repressilator experiment. This figure coincides with the one in the main text, but here we provide it with

the training data evolving over time steps. It is clear from the figure that our method can use the information from all samples to find an accurate global fit, as clearly shown by the visual similarity to the ground truth and lower mean squared error on the lattice (0.807). The global-forward-SB method, on the other hand, predicts a vector field of smaller magnitude, potentially because the reference measure serves as a regularization, and the mean squared error is higher (9.998). In fig. 3 we also show that the vanilla SB approach fails to capture the global dynamics: for each pair of time steps, the vanilla SB approach provides a very local vector field around the marginal samples at those two time steps. Even around marginal samples, though, the predictions are not accurate. This is expected, as this approach does not have access to the global dynamics and is therefore missing the big picture. In terms of MSEs, for the four pairs of time steps that we consider, the vanilla SB method achieves 17.8, 17.2, 16.7 and 16.6, respectively.

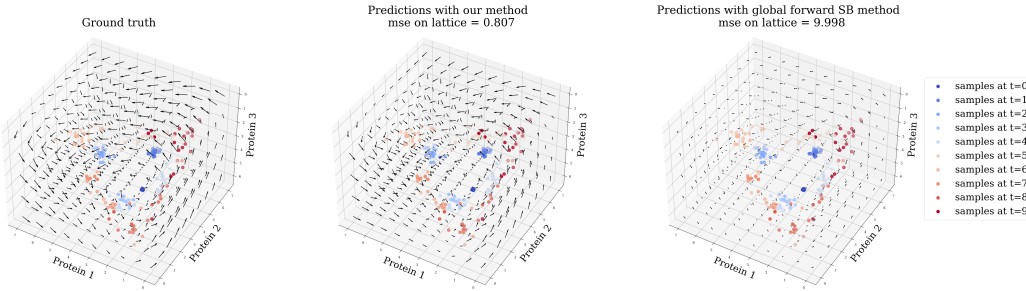

Figure 2: Vector field reconstruction in the repressilator experiment. Left: the ground truth vector field generating data. Mid: reconstruction using our method. Right: reconstruction using global forward SB method. Our method's MSE on the lattice is 0.807, the alternative's MSE is 9.9.

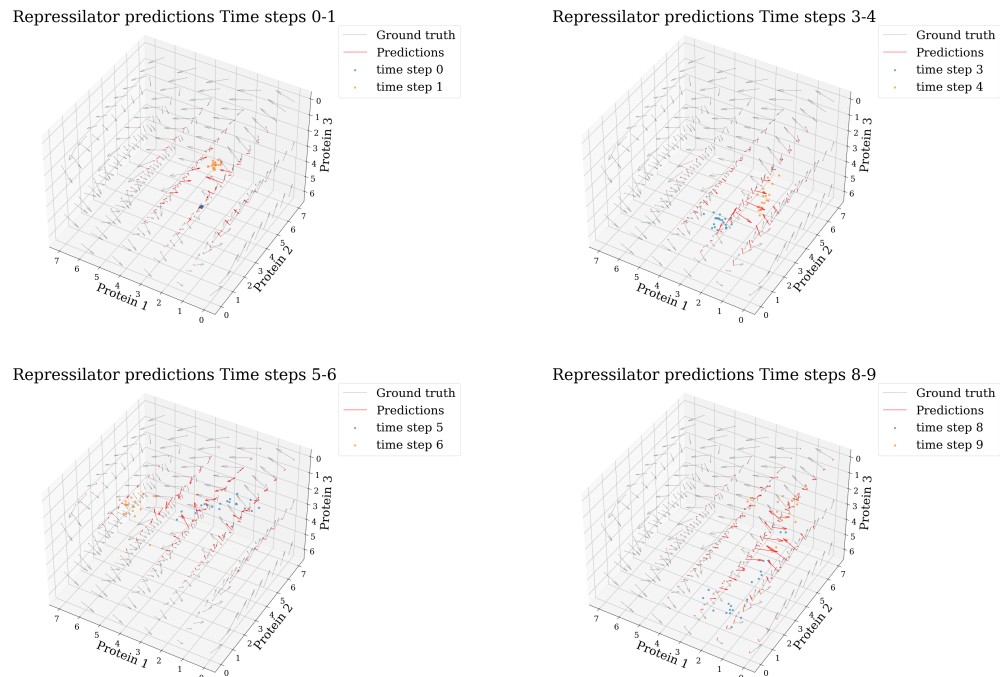

Figure 3: Vector field reconstruction for the repressilator experiment using vanilla SB. Each panel shows the vector field reconstruction in between two different time steps. It is clear that the vanilla SB approach fails to capture the global dynamics, since it does not have access to samples besides the two time steps of interest. This is reflected in (1) poor predictions in between those two time steps (since it does not leverage information coming from other time points), and (2) poor generalization outside those two time steps (the reconstructed field is very close to zero). MSEs for these four learnt fields are 17.8, 17.2, 16.7 and 16.6.

### E.2 LOTKA-VOLTERRA

**Experiment setup.** For this experiment, we are interested in learning the dynamics of a stochastic Lotka-Volterra predator-prey model. The dynamics of the prey and predator populations are given by the following SDEs:

$$dX = \alpha X - \beta XY + 0.1 dW_x$$
$$dY = \gamma XY - \delta Y + 0.1 dW_y$$

where $[dW_x, dW_y]$ is a 2D Brownian motion. To obtain data, we fix the following parameters: $\alpha = 1, \beta = 0.4, \gamma = 0.1, \delta = 0.4$.

We start the dynamics at $X_0 \sim U(5, 5.1)$ and $Y_0 \sim U(4, 4.1)$, and simulate the SDEs for 10 instants of time. At each time step, we take 20 samples. We use Euler-Maruyama method to obtain the numeric solutions.

**Results.** We show the results in fig. 4. Our method reconstructs very accurately the underlying dynamics, with a mean squared error of 0.05. The global-forward-SB method also produces a predicted vector field which is very similar to the ground truth, but with a higher mean squared error (1.265). For each pair of time steps, the vanilla SB approach provides a very local vector field around the marginal samples at those two time steps. We show the vector field reconstruction for four pairs of time steps using the vanilla SB in fig. 5.

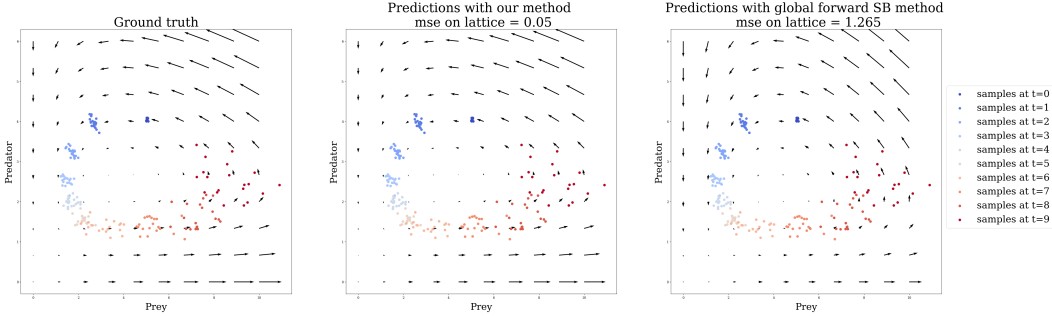

Figure 4: Vector field reconstruction for the Lotka-Volterra experiment. Left: the ground truth vector field generating data and the training data evolving over time steps. Mid: reconstruction using our proposed method. Right: reconstruction using SB method with a global forward drift. Our method achieved an MSE on the grid 0.05 while the alternative achieved 1.265.

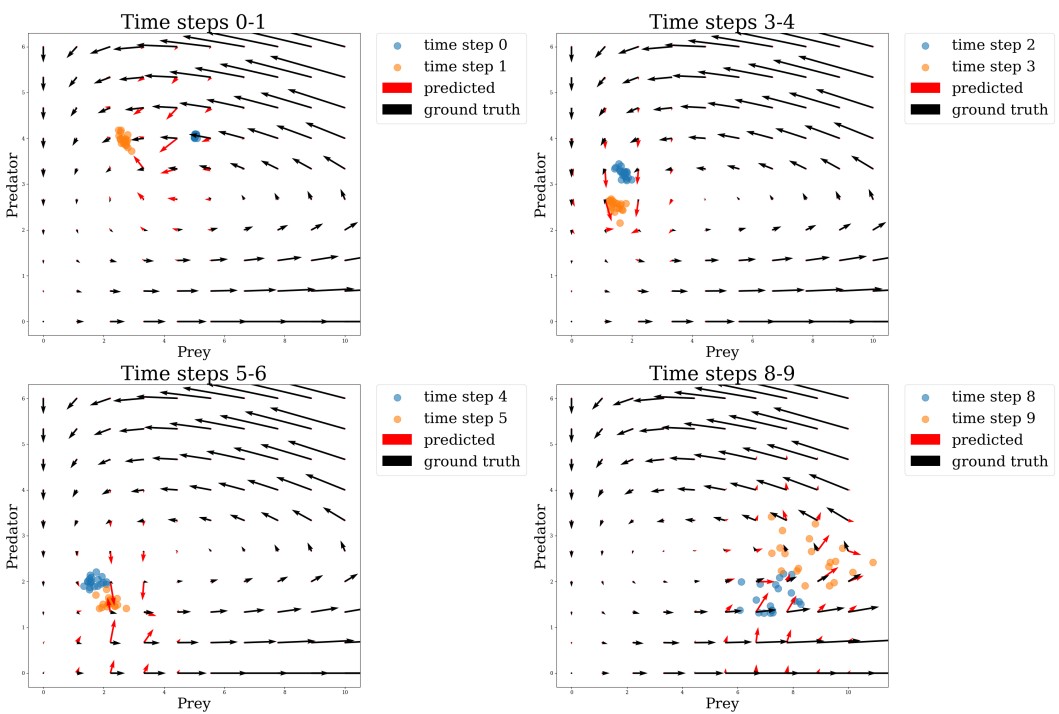

Figure 5: Vector field reconstruction for the Lotka-Volterra experiment using vanilla SB. Each panel shows the vector field reconstruction in between two different time steps. The vanilla SB approach fails to capture the global dynamics. MSEs for these four learnt fields are 11.4, 11.8, 11.8 and 11.5, respectively.

