# OpenReview forum: "Learning a vector field from snapshots of unidentified particles rather than particle trajectories"
_ICLR.cc/2024/Workshop/AI4DiffEqtnsInSci — AI4DiffEqtnsInSci @ ICLR 2024 Poster_

### Official Review · Reviewer_MjN1 · 2024-02-25
**This is a novel and interesting study**

**Rating:** 9
**Confidence:** 3

**Review:**

This paper presents a novel approach to infer the dynamics of systems from sparse time snapshots without access to individual trajectories. Using an iterative projection mechanism inspired by Schrödinger bridges, the authors propose a method that iteratively refines a dynamical model to match observed marginal distributions over multiple snapshots. This approach is tested on simulated data from ecology, demonstrating its potential to adequately reconstruct the underlying dynamics.

Pros:
- The paper aims to address a significant challenge in dynamical systems, especially in areas where only marginal snapshots are available.
- The work may be of interest and relevance to a broad audience in the workshop, with the new tool for understanding dynamical systems potentially applicable and useful to other fields.

Cons:
- The robustness of the methodology to data sets with different characteristics (e.g., data sparsity, and underlying system dynamics with different parameters, as well as different levels of noise as mentioned by the authors) is not extensively explored.

Comments/Questions:
- The choice of experiments in ecology may seem disconnected from the initial example focused on mRNA dynamics, which confuses me as I expect a direct application of the proposed method to the scenario described in Sec 2. I would suggest that the paper clarify the rationale behind the choice of actual experiments and explain how they relate to the broader goals of the research.
- I question certain assumptions about the sparsity and distribution of data points on which the methodology is based. Please consider clarifying these.
- Does the method require some parameters such as the choice of reference dynamics or optimization settings, and I wonder if there is some exploration of sensitivity analysis for the parameter selection.
- What are the computational requirements for the proposed method and what is the efficiency or scalability of the algorithm compared to the baseline method?

---

### Meta-Review · Area_Chair_8Yns · 2024-03-01

**Recommendation:** Accept (Poster)

**Metareview:**

This paper presents a new method that get inspired by Schrödinger bridge, which can be used for inferring the dynamics of systems. The study is comprehensive despite it's only 4 page. However, author should address reviewer MjN1's comments and questions in the camera ready version

---

### Decision · Program_Chairs · 2024-03-02

Accept (Poster)